# Efficacy of Infection Eradication in Antibiotic Cement-Coated Intramedullary Nails for Fracture-Related Infections, Nonunions, and Fusions

**DOI:** 10.3390/antibiotics11060709

**Published:** 2022-05-25

**Authors:** Janet D. Conway, Ahmed H. Elhessy, Selin Galiboglu, Nirav Patel, Martin G. Gesheff

**Affiliations:** International Center for Limb Lengthening, Rubin Institute for Advanced Orthopedics, Sinai Hospital of Baltimore, Baltimore, MD 21215, USA; ahmed_hessy@hotmail.com (A.H.E.); sgaliboglu@lifebridgehealth.org (S.G.); nirpatel@lifebridgehealth.org (N.P.); mgesheff@lifebridgehealth.org (M.G.G.)

**Keywords:** antibiotic cement-coated intramedullary nails, ACCIN, FRI, osteomyelitis, infected arthrodesis, infected fusion, infected nonunion

## Abstract

Antibiotic cement-coated intramedullary nails (ACCINs) are increasing in popularity as a viable solution for the treatment of fracture-related infections (FRIs), infected long bone nonunions, and arthrodeses without an external fixator. ACCINs effectively manage to fulfill three of the basic principles for eradicating osteomyelitis: dead space management, antibiotic delivery, and bone stability. We performed a retrospective review of 111 patients who were treated with ACCINs between January 2014 and December 2020. In our series, 87.4% (*n* = 97) of patients achieved healed and uninfected bone or stable arthrodesis at a mean follow-up of 29.2 months (range, 6–93 months). Additionally, 69.1% (*n* = 67) of healed patients were resolved after only one procedure, and the remainder (30.9%, *n* = 30) healed after one or more additional procedures. The mean number of additional procedures was 2.1 (range, 1–6 additional procedures). The overall limb salvage rate was 93.7% (*n* = 104). The majority of the total cohort were successfully treated in only one surgery. This study suggests that ACCINs are effective in the treatment of FRIs, infected long bone nonunions, and infected ankle and knee arthrodeses.

## 1. Introduction

Orthopedic infections complicated by large soft tissue defects, exposed hardware, and lack of bone stability have traditionally been treated with external fixation since inserting internal fixation into an infected bone and soft tissue envelope has been contraindicated. Bone stability is essential for eradicating infection. The concept of the insertion of antibiotic-coated intramedullary nails has revolutionized the care of these difficult cases since it can provide antibiotic delivery and bone stability and be inserted at the time of radical infection debridement.

Antibiotic cement-coated intramedullary nails (ACCINs) allow for early weight bearing, prompt joint range of motion, and the possibility of treatment in a single operation. First described in 2007, ACCINs offer the advantages of providing both antibiotic delivery and bone stability [1]. The literature shows multiple studies involving ACCINs that have demonstrated their successful implementation to treat infected nonunions and arthrodeses [2,3,4,5,6]. The 2018 International Consensus Meeting on Musculoskeletal Infection agreed that antibiotic-coated intramedullary nails were an effective treatment for these difficult cases [7]. Additionally, an international expert group on fracture-related infection (FRI) published evidence-based recommendations for local antibiotic delivery and dead space management, and declared that the use of intramedullary antibiotic-laden polymethyl methacrylate (PMMA)-coated nails remains a good treatment option for long bone FRIs [8].

In this present study, 111 patients who underwent procedure with ACCINs from January 2014 to December 2020 at a single center by a single surgeon were retrospectively investigated. This is the largest known series of patients undergoing this ACCIN technique, and the most recent group investigated by the senior author. These findings continue to demonstrate the ACCINs efficacy with respect to infection control, now demonstrated for an increased number of indications.

## 2. Results

In our series, 87.4% (*n* = 97) of patients achieved either healed and uninfected bone or stable arthrodesis, as applicable, at a mean follow-up of 29.2 months (range, 6–93 months). This was confirmed clinically, radiographically, and serologically. Of this subset of 97 patients, 69.1% (*n* = 67) healed after only one procedure and 30.9% (*n* = 30) after one or more additional procedures. The mean number of additional procedures was 2.1 (range, 1–6 procedures). Secondary procedures were performed at a mean time interval of 10.5 months (range, 1–45 months). Of the remaining 14 patients, seven ended with amputation and seven were still undergoing treatment: three for infected nonunion and four for nonunion only after infection resolution.

Thirty patients who were successfully treated underwent additional procedures. Fourteen of them had the same culture results postoperatively and the remaining 16 had different results postoperatively. Twenty patients had positive culture results and the remaining 10 had negative culture results. Regarding antibiotics susceptibility for the 20 patients with positive culture results, 18 patients were sensitive to several antibiotics including vancomycin or tobramycin. One patient (with multiple organisms) was only sensitive to meropenem and was resistant to both tobramycin and vancomycin. That patient received meropenem postoperative for six weeks. His infection was resolved by completion of his follow-up. The remaining patient in this subset of 20 patients with positive culture results did not have an antibiotic susceptibility report available in their records. Of the 30 patients who required additional procedures, 24 patients (80%) required an additional procedure due only to infection, and the mean number of procedures was two (range, 1–5 procedures). The most common procedures for infection were irrigation and debridement, bone resection, exchange rodding, and rod removal with antibiotic calcium sulfate injection.

The mean time from the initial ACCIN to the additional procedure was 7.6 months (range, 1–29 months). Four patients (13.3%) required additional procedures due only to nonunion, and the mean number of procedures was 1.3 (range, 1–2 procedures). In all nonunion cases, the procedure was exchange rodding with bone morphogenetic protein (BMP)-2 and/or reamer-irrigator-aspirator (RIA) autograft bone. The mean time from the initial ACCIN to the additional procedure was 18.3 months (range, 3–45 months). The last two patients (6.7%) needed additional procedures for both infection and nonunion. Both patients required only one additional procedure: a bone resection with exchange rodding and insertion of BMP-2. The mean time from the initial ACCIN to the additional procedure was 2.5 months (range, 1–4 months).

Thirty-five patients had segmental bone defects; mean defect size was 6.1 cm (range, 0.5–24 cm). The success rate within this subgroup was 85.7% (*n* = 30). The success rate for patients without defects was 88.2% (*n* = 67). There was no significant difference between the number of additional procedures needed for success in both groups.

Only seven patients required an amputation; the overall limb salvage rate was 93.7%. The mean age for patients who underwent amputation was 53.7 years (range, 44–61 years). Five patients underwent above-knee amputation and the remaining two had below-knee amputation. The mean number of additional procedures that were performed after the initial ACCIN and before amputation was 1.1 (range, 0–3 procedures). The mean time from first ACCIN until amputation was 30.5 months (range, 15–50 months). Four amputation patients had negative cultures and three were positive for Staphylococcus aureus. Five amputations were performed due to persistent infection and two were due to nonunion.

Regarding complications, 22 patients presented with 25 complications directly related to the initial insertion of the ACCIN. Symptomatic hardware was the most common (40%, *n* = 10), followed by superficial infection (24%, *n* = 6), nerve compression (20%, *n* = 5), joint contracture (8%, *n* = 2), and finally, hematoma and broken hardware, each occurring once (4% each) (Table 1). The lone broken nail was the result of a knee fusion ACCIN for treatment of an infected total knee arthroplasty (TKA) with a residual bone defect of 9 cm. This case was revised with another knee fusion nail and bone transport over the nail using a monolateral fixator, resulting in successful noninfected fusion. Superficial infections were handled with local wound care and without a return to the operation room. Superficial infection happened in the first 2 weeks after surgery. All patients were already on postoperative antibiotic treatment for a total of 6 weeks.

Fifty patients (45%) of the series underwent rod removal for various reasons, with a total of 66 rods removed. The reasons for ACCIN removal were recurrence of infection (43.9%, *n* = 29); as a second stage of TKA revision (24.2%, *n* = 16); symptomatic hardware (15.2%, *n* = 10); as a part of an amputation procedure (10.6%, *n* = 7); and the remaining removals were due to nonunion, planning, and a broken rod (Table 2). Debonding occurred during ACCIN removal in only four cases.

A case example of a 74-year-old male patient who underwent ankle fusion using an ACCIN after an open ankle fracture is available, with several radiographs and photos, in the Appendix A.

## 3. Discussion

ACCINs continue to gain attention as a viable solution for the treatment of FRIs, infected long bone nonunions, and arthrodeses without an external fixator [9]. The basic principles for eradicating osteomyelitis include bone debridement, dead space management, antibiotic delivery, and bone stability. ACCINs can effectively manage to fulfill the latter three principles listed above. An additional advantage to this technique is that immediate weight bearing is possible.

Since 2007, many authors have published moderately sized series describing success using this technique [5,6]. Karek et al. [10] compared PMMA antibiotic elution using chest tube-coated guidewires and intramedullary nails. They found that the coating on the intramedullary nails had greater elution despite the thinner cement mantle. At least 50% of the antibiotics eluded in the first 24 h and the remaining antibiotic elution was at a sufficient level to inhibit *S. aureus* for 6 weeks [10]. Lopas et al. [11] described 41 cases of septic long bone nonunion and compared 33 flexible nonlocked rods to eight ACCINs. They reported that patients treated with ACCINs achieved earlier weight bearing, required autograft less frequently, and underwent fewer subsequent procedures when compared to the patients with flexible nails. In their retrospective study, Makhdom and his colleagues concluded that ACCINs represent an effective treatment option for septic complex lower extremity reconstruction and are associated with a limb salvage rate of 89%. They also recommended that patients with knee fusion after failed TKA should be counseled because of a high potential complication rate [3].

There are variations in the technique of treating infection with ACCINs, which is due to the various ways that an intramedullary nail can be coated. One article describes hand-rolling the nails [12], whereas other publications use chest tubes or vascular perfusion tubing [2,5]. In the report by Rice et al. [5] on single-stage treatment of FRIs, ACCINs were created using vascular perfusion tubing and resulted in 69% (22/32) of patients with union and were free from infection after one procedure with the ACCIN. Those patients who required additional surgery for infection or nonunion averaged 1.9 additional procedures per patient [5].

All cases in the present study used the silicone tubing technique, as previously published in a review that included patients treated with that technique as well as the mold technique [13]. In that report, 57% (38/67) of infected arthrodesis cases were healed and noninfected, and 60% (26/43) of infected long bone nonunion cases were treated successfully with one surgery with ACCIN. The present series used only the silicone tubing technique. This choice has greatly facilitated the ease with which the rod is created and has essentially eliminated the problem of debonding upon insertion or removal. The rate of ACCIN use at the authors’ institution has substantially increased, as have the number of cases seen at our referral center.

Segmental bone defects can pose a considerable challenge. In our previous series [13], the segmental bone defect group required more surgeries than those without segmental defects. In the present series, there was no statistically significant difference in the number of additional surgeries between segmental and non-segmental groups. Although the size of the defect significantly negatively impacted the success rate of the ACCIN and the number of additional procedures required to achieve a successful result in our 2014 study, it was not the case in the present series. The reason for this is unclear, considering the mean segmental defect size was 6 cm. Other literature similarly indicates that larger segmental defects are more difficult. In a study of 25 patients, Shyam et al. [12] reported that all patients with >6 cm segmental defects failed to heal and eradicate infection. The success rate in our segmental defects patients was 85.7% with three patients undergoing amputation.

Hardware removal after ACCIN insertion can be done for variable reasons and has been discussed in literature [14]. The most common reason for removal of rods in our series was recurrent infection (Table 2). Reasons for removal, pearls, and complications were previously detailed in our 2021 work [14].

This study has limitations, including its retrospective nature by a single surgeon. Additional limitations are the absence of a comparative group, the variety of indications, and the presence of segmental bone defects in a subgroup of the series. Additionally, the variation of the age group (range, 13–83 years) may have influenced the outcome. However, the authors posit that the variety of indications and the inclusion of segmental bone defects promotes the effectiveness of ACCINs in the treatment of bone infection, even in complex cases.

This study demonstrates the effectiveness of the ACCIN technique in the setting of infected nonunions and arthrodeses. The silicone tubing method is effective for creating a uniform cement mantle that can withstand insertion and removal. Several studies on commercially available European ACCINs have shown promise in treating open fractures and tibial nonunions [15,16,17,18] and expanding their use in the U.S. will make it more feasible for larger multicenter trials that use this technique.

## 4. Materials and Methods

After obtaining Institutional Review Board (IRB) approval, a retrospective nonrandomized chart review was performed on a series of patients from January 2014 to December 2020 who were identified from a single surgeon database at a single institution. The investigation was performed in accordance with the principles of the Declaration of Helsinki. Within the specified timeframe, all ACCINs inserted by a single surgeon for the treatment of infected nonunions and arthrodeses were evaluated. The review revealed 120 patients who received ACCINs within that time. Patients with a follow-up of <6 months and patients who were classified as type C hosts according to the Cierny–Mader classification were excluded [19]. The present study is comprised of the 111 patients who remained after exclusions.

Demographics of our cohort are detailed in Table 3 and were as follows: there were 57 (51.4%) males and 54 (48.6%) females with a mean age of 56.9 years (range, 13–83 years). The mean body mass index (BMI) was documented for all patients on the day of procedure and was 33.4 kg/m^2^ (range, 17.5–52.4 kg/m^2^). Five patients (4.5%) were type A hosts and 106 patients (95.5%) were type B hosts. Diabetes and smoking were the most common comorbidities. Regarding race, 66.7% (*n* = 74) were white; 28.8% (*n* = 32) African American; 1.8% (*n* = 2) others, which included Asians and Native Hawaiian; and 2.7% (*n* = 3) declined to answer. Indications for the use of an ACCIN were as follows: infected TKAs (38.7%, *n* = 43), infected fusions (20.7%, *n* = 23), FRIs (20.7%, *n* = 23), and infected nonunions (19.8%, *n* = 22). Nails used were knee arthrodesis (46%, *n* = 51), ankle arthrodesis (32.4%, *n* = 36), tibial (11.7%, *n* = 13), retrograde femoral (5.4%, *n* = 6), and antegrade femoral (4.5%, *n* = 5).

Segmental bone defects were measured intraoperatively after debridement and were defined as the smallest-sized bone defect that would not heal without intervention [20]. Patients who had segmental bone defects represent 31.5% (*n* = 35) of the total cohort, with a mean defect size of 6.1 cm (range, 0.5–24 cm). This subgroup was divided into patients with infected total joints comprised 19.8% (*n* = 22) with a mean defect size of 6.7 cm (range, 1–24 cm) and patients with other causes who comprised 11.9% (*n* = 13) with a mean defect size of 5.2 cm (range, 0.5–12 cm) (Table 3). Specifically, the other causes consisted of 5 infected nonunions, 6 infected fusions, and 2 FRIs.

Full culture results are listed in Table 4. The most common organisms were methicillin-susceptible *S. aureus* (MSSA) in 17.1% (*n* = 19) of patients and methicillin-resistant *S. aureus* (MRSA) totaling 12.6% (*n* = 14). Negative wound cultures were reported in 42.3% (*n* = 47) of patients and 12.6% (*n* = 14) had multiple organisms. Cultures were obtained for the four patients with nonunion only; their results were negative. The same criteria were followed for these 4 to confirm there was no infection and the revision was only for nonunion. Negative culture results can be associated with osteomyelitis; thus, diagnosis involves clinical, laboratory, and radiological investigations. Some studies have suggested that the rate of negative cultures obtained during surgery in histologically proven osteomyelitis exceeds 40% [21]. Antibiotics were not routinely discontinued prior to surgery and could account for our culture rate. However, up to a 30% negative culture rate has been reported in the literature [22].

Although FRI is a challenging musculoskeletal complication in trauma surgery, it long lacked a clear definition. In their efforts to provide a clear definition for FRI, Metsemakers et al. suggested that although acute vs. chronic infections mandate different treatment strategies, this should not impact how clinicians define FRI [8]. In our series, the senior author classified FRIs as acute fractures <6 months and infected nonunions as those presenting >6 months, whereas the US Food and Drug Administration’s classification of a nonunion is those presenting ≥9 months without showing healing for 3 consecutive months. Infection was defined using the criteria from the Musculoskeletal Infection Society (with the presence of a draining sinus, erythema or abscess clinically, radiographically with osteolysis on radiographs or bone changes on MRI consistent with infection, positive intraoperative biopsies, and elevated serologic markers [C-reactive protein and erythrocyte sedimentation rate]) in case of PJI [23]. In case of FRI, we followed the same recommendations that were recently published to diagnose FRI [24]. Preoperative management prior to surgery included patient optimization such as correcting preoperative vitamin D deficiency, blood glucose levels, smoking cessation, and vascular evaluation when the pulses were not symmetric.

The surgical technique was standardized, and all surgeries were performed by the senior surgeon. Thorough debridement was performed, and cultures were taken at the initial surgery. All the nails used in this study were the same model from the same manufacturer (Trigen; Smith & Nephew, Memphis, TN, USA). These standard intramedullary nails were coated using silicone tubing of either 11 mm or 12.5 mm diameter and Palacos cement (Heraeus Medical, Hanau, Germany) containing 1 gm of vancomycin and 3.6 gm of tobramycin per 40 gm of cement (Figure 1). The rods were then placed and locked under fluoroscopic guidance. The surgical technique has been previously described [1,13].

Postoperatively, 6 weeks of organism-specific antibiotics were given. In culture-negative infections, broad-spectrum antibiotics were used. When possible, in the culture-negative infections, historical cultures were used to guide antibiotic therapy. Antibiotic management was directed by the infectious disease consultant. The combination of local vancomycin and tobramycin is favored in the treatment of osteomyelitis [25]. Chemical prophylaxis for venous thromboembolus was used for 6 weeks. Postoperative weight bearing was initiated as soon as possible, based upon the size of the bone defect. All patients who underwent knee arthrodesis were fully weight bearing immediately postoperative, even with segmental defects, because of the stability of the intercalary antibiotic cement spacer. All patients who underwent ankle arthrodesis were non-weight bearing for 6 weeks and then transitioned to a Charcot restraint orthotic walker (CROW) boot for full weight bearing. Range of motion was initiated immediately in all long bone nonunions and weight bearing was determined on an individual basis, dependent upon the size of the segmental defect.

All patients were monitored for resolution of infection and bone healing. Serial C-reactive protein and erythrocyte sedimentation rate values were obtained weekly for 6 weeks, followed by testing every 2 weeks to ensure that lab values returned to normal and stayed normalized following completion of 6 weeks of antibiotic therapy. Clinical monitoring of infection was performed routinely at 2, 4, 8, and 12 weeks postoperatively. Radiographic evaluation was performed to assess for bone healing at 4, 8, and 12 weeks postoperatively. If there was an equivocal radiographic evaluation at 12 weeks, a computed tomography scan was performed at 16 weeks to assess for bone bridging and healing of the long bone nonunion. Bone union was determined to be achieved when 3 of 4 cortices were united and the patient was fully weight bearing without pain. In patients with intercalary spacers and knee arthrodesis rods, bone union was not always the goal. There was often no further surgery indicated for medically debilitated patients. The remaining patients had either formal fusion with removal of the intercalary spacer or otherwise converted to revision TKA. A workflow diagram is available in Figure 2.

After removing identifying patient information, the data were recorded in Excel (Microsoft, Redmond, WA, USA). All statistical analyses were performed using MedCalc statistical software (version 20.009; MedCalc Software Ltd.; Ostend, Belgium). Demographics were documented using mean with standard deviation and percentages of the entire study population. Comparisons of proportions were compared with a chi-squared test or Fisher exact test for comparison of <5 cases per cell.

## 5. Conclusions

ACCINs are an effective technique for treating FRIs and infected long bone nonunions, as well as infected arthrodeses of the knee and ankle. In our review of 111 patients the overall limb salvage rate was 93.7%, and the majority of patients were successfully treated with a single surgical procedure. Future multicenter randomized studies comparing the outcome of ACCINs versus other modalities in the treatment of bone infection are encouraged.

## Figures and Tables

**Figure 1 antibiotics-11-00709-f001:**
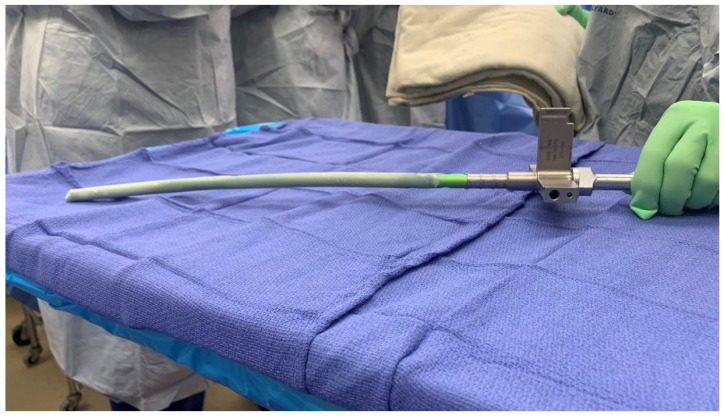
Photo of an antibiotic-coated hindfoot fusion rod (Trigen; Smith & Nephew, Memphis, TN, USA). The width of the nail with cement is 12 mm.

**Figure 2 antibiotics-11-00709-f002:**
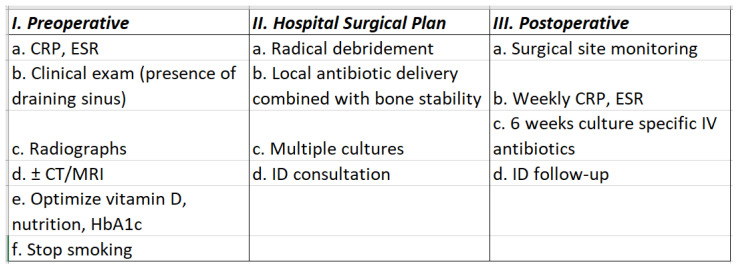
Workflow diagram. CRP, C-reactive protein; CT, computed tomography; ESR, erythrocyte sedimentation rate; HbA1c, blood sugar; ID, infectious diseases; IV, intravenous; MRI, magnetic resonance imaging.

**Table 1 antibiotics-11-00709-t001:** Complications.

Complication	*n* (%)	Treatment
Symptomatic hardware	10 (40%)	Removal
Superficial infection	6 (24%)	Intravenous antibiotics ± local wound care
Nerve compression	5 (20%)	Decompression
Joint contracture	2 (8%)	Soft-tissue release
Hematoma	1 (4%)	Irrigation and debridement with drain insertion
Broken hardware	1 (4%)	Revised with another fusion nail

**Table 2 antibiotics-11-00709-t002:** Rod Removal.

Reason	*n* (%)
Recurrent infection	29 (43.9%)
2nd stage of TKA	16 (24.2%)
Symptomatic hardware	10 (15.2%)
Amputation	7 (10.6%)
Nonunion	2 (3%)
Planned by surgeon	1 (1.5%)
Broken hardware	1 (1.5%)

TKA, total knee arthroplasty.

**Table 3 antibiotics-11-00709-t003:** Demographics.

Demographic	*n* (%)	
Male	57 (51.4)	
Female	54 (48.6)	
Age	56.9 years (range, 13–83 years)	
BMI	33.4 (range, 17.5–52.4 kg/m^2^)	
Follow-up	29.2 months (range, 6–93 months)	
Host type A	5 (4.5)	
Host type B	106 (95.5)	
**Preoperative diagnosis**		
Infected total joint	43 (38.7)	
Infected fusion	23 (20.7)	
Fracture-related infection	23 (20.7)	
Infected nonunion	22 (19.8)	
**ACCIN locations**		
Knee fusion nail	51 (46)	
Hindfoot fusion	36 (32.4)	
Tibia nail	13 (11.7)	
Femur retrograde nail	6 (5.4)	
Femur antegrade nail	5 (4.5)	
Patients with segmental defects, total	35 (31.5)	Defect size = 6.1 cm (range, 0.5–24 cm)
Infected TKA	22 (19.8)	Defect size = 6.7 cm (range, 1–24 cm)
Others	13 (11.7)	Defect size = 5.2 cm (range, 0.5–12 cm)

ACCIN, antibiotic cement-coated interlocking nail; BMI, body mass index; TKA, total knee arthroplasty.

**Table 4 antibiotics-11-00709-t004:** Microbiological Cultures.

Culture	*n* (%)
MSSA	19 (17.1)
Multiple organisms	14 (12.6)
MRSA	15 (13.5)
*Pseudomonas aeruginosa*	4 (3.6)
*Enterobacter cloacae*	4 (3.6)
*Corynebacterium*	3 (2.7)
*Streptococcus agalactiae*	2 (1.8)
*Proteus mirabilis*	1 (0.9)
*Enterococcus faecalis*	1 (0.9)
*Candida albicans*	1 (0.9)
Negative	47 (42.3)

MRSA, methicillin-resistant *Staphylococcus aureus*; MSSA, methicillin-susceptible *S. aureus*.

## Data Availability

The data presented in this study are available upon request from the corresponding author. The data are not publicly available due to privacy concerns with protected health information.

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
