# Peer review of "Efficacy of Infection Eradication in Antibiotic Cement-Coated Intramedullary Nails for Fracture-Related Infections, Nonunions, and Fusions"

_antibiotics, 2022, doi:10.3390/antibiotics11060709_

Round 1
Reviewer 1 Report
The authors should consider the followings:
- The authors are advised to add workflow diagrams to the article, for the surgical procedures and for the clinical infection monitoring (with the dose and types of antibiotics, if any), respectively.
- Since body weights and/or diabetic conditions may be influential to the overall infection status. The authors may provide further information as follows: i. Were the subjects monitored with their diabetes conditions? ii. Please list the diabetes condition in the form of demographics. iii. Were the BMI documented, before and after the surgery? iv. Please supplement this data (iii) to the research article.
- In Table 3 demographics, please list the % ethnicity of the subjects.
- The authors may provide the information (demographic or case-wise), of CRP and ESR lab test (as refer to line 218).
- The authors may revise the title of Table 4, as microbiological cultures. Please include the standardized scientific name of the tested microbial, in the Table 4.
- Please mark a scale bar to the photo of Figure 1 and to those Supplementary Figures.
- In the part of Conflicts of Interest, at line 300, would JDC have further financial disclosures to report?
- Please check the spelling of “Palicos” cement, if spelt correctly.
- In section 4 (Materials and Methods), please supplement the review case number of the Institutional Review Board.
- The authors should clearly mention the novel findings of the research, in the part of abstract and conclusion.
Author Response
Reviewer #1
The authors should consider the followings:
- The authors are advised to add workflow diagrams to the article, for the surgical procedures and for the clinical infection monitoring (with the dose and types of antibiotics, if any), respectively.
Response: We have created this image and legend and inserted them, along with a callout sentence at line 299:
Fig 2. Workflow diagram. CRP, C-reactive protein; CT, computed tomography; ESR, erythrocyte sedimentation rate; HbA1c, blood sugar; ID, infectious diseases; IV, intravenous; MRI, magnetic resonance imaging.
- Since body weights and/or diabetic conditions may be influential to the overall infection status. The authors may provide further information as follows: i. Were the subjects monitored with their diabetes conditions? ii. Please list the diabetes condition in the form of demographics. iii. Were the BMI documented, before and after the surgery? iv. Please supplement this data (iii) to the research article.
Response: Some patients were diabetic. We chose to report the host type according to the Cierny-Mader classification. BMI was documented for all patients on the day of the procedure. This sentence was added at line 209.
- In Table 3 demographics, please list the % ethnicity of the subjects.
Response: We have added this text in the demographics paragraph at lines 212-214. Regarding race, 66.7% (n=74) were white, 28.8% (n= 32) African American, 1.8% (n=2) others which included Asians and Native Hawaiian, and 2.7% (n=3) declined to answer
- The authors may provide the information (demographic or case-wise), of CRP and ESR lab test (as refer to line 218).
Response: Done as a part of the routine preoperative workup for any infected patient to aid the diagnosis and for the sake of follow-up.
Lines 252-256: (diagnosis)
Infection was defined using the criteria from the Musculoskeletal Infection Society (with the presence of a draining sinus, erythema or abscess clinically, radiographically with osteolysis on radiographs or bone changes on MRI consistent with infection, positive intraoperative biopsies and elevated serologic markers [C-reactive protein and erythrocyte sedimentation rate])
Lines 287-290: (follow-up)
All patients were monitored for resolution of infection and bone healing. Serial C-reactive protein and erythrocyte sedimentation rate values were obtained weekly for 6 weeks, followed by testing every 2 weeks to ensure that lab values returned to normal and stayed normalized following completion of 6 weeks of antibiotic therapy.
- The authors may revise the title of Table 4, as microbiological cultures. Please include the standardized scientific name of the tested microbial, in the Table 4.
Response: Thank you, we have revised the title of the table. We have also updated the names of the microbials.
- Please mark a scale bar to the photo of Figure 1 and to those Supplementary Figures.
Response: Our multimedia specialist advised that it is not feasible to add a scale bar to the photographs at this point. We are not familiar with this request in our past experience. We did add to the figure 1 legend that “the width of the nail with cement is 12 mm.” This is at line 270.
- In the part of Conflicts of Interest, at line 300, would JDC have further financial disclosures to report?
Response: The conflicts of interest are complete as written.
- Please check the spelling of “Palicos” cement, if spelt correctly.
Response: Updated to Palacos at line 265, thank you for pointing this out.
- In section 4 (Materials and Methods), please supplement the review case number of the Institutional Review Board.
Response: No case number; LifeBridge Health IRB determined this to be exempt on March 24, 2021 based on applicable federal regulations 45 CFR 46.104(d)(2 and 4). .
- The authors should clearly mention the novel findings of the research, in the part of abstract and conclusion.
Response: We feel these are clearly mentioned in the abstract and conclusion.

Reviewer 2 Report
- The English need improvement since there are some grammatical and syntax errors in the manuscript. For example,
- in line number 44, the word “procedure” may be as “the procedure”;
- in line number 58 and 63, “and mean” as “and the mean”;
- in line number 65, “Mean” as “The mean”;
- in line number 80, “first” as “the first”;
- in line number 127, “infection” as “the infection”;
- in line number 130, “single-stage” as “the single-stage”;
- in line number 131, “union” as “the union”;
- in line number 158, “in literature” as “in the literature”;
- in line number 259, “removal” as “the removal”.
The grammar mistakes which are not mentioned here are also to be checked and corrected properly.
- There are some typing mistakes as well, and authors are advised to carefully proofread the text. For example,
- all over the manuscript, the word “weightbearing” may be as “weight bearing”;
- in line number 42, “remain” as “remains”;
- in line number 61, “Mean time” as “Meantime”;
- in line number 234, “organism specific” as “organism-specific”;
- in line number 235, “broad spectrum” as “broad-spectrum”.
The typos not mentioned here are also to be checked and corrected properly.
- Check the abbreviations throughout the manuscript and introduce the abbreviation when the full word appears the first time in the text and then use only the abbreviation (For example, fracture-related infections (FRI), BMP-2, etc.). And it should be in both abstract as well as in the remaining part of the manuscript. Make a word abbreviated in the article that is repeated at least three times in the text, not all words need to be abbreviated.
- The full form of the species should be given when the first time appears in both abstract and in the remaining part of he manuscript and it should be followed by only the first letter of the genus (e.g., Staphylococcus aureus when the first time appears and followed by S. aureus). And it should be for all other species used.
- The intrudouction part appears less informative about the orthopaedic infection, thus this section should be indicated as detailed to understand the manuscript in clear.
- In table 2, the words used may start with an upper case letter (capital letter).
- The authors may improve the discussion of their work by focusing on the present findings and introducing other authors who also worked with the same or other studies with recent references shortly.
8. In conclusion, it is highly recommended to include weaknesses of the study and potential future research goals.
Author Response
Reviewer #2
- The English need improvement since there are some grammatical and syntax errors in the manuscript. For example,
- in line number 44, the word “procedure” may be as “the procedure”;
- in line number 58 and 63, “and mean” as “and the mean”;
- in line number 65, “Mean” as “The mean”;
- in line number 80, “first” as “the first”;
- in line number 127, “infection” as “the infection”;
- in line number 130, “single-stage” as “the single-stage”;
- in line number 131, “union” as “the union”;
- in line number 158, “in literature” as “in the literature”;
- in line number 259, “removal” as “the removal”.
The grammar mistakes which are not mentioned here are also to be checked and corrected properly.
Response: We felt that dropping the definite article “the” in these instances aided in making our writing concise, and that this stylistic practice is common and akin to published medical research articles found in Antibiotics and elsewhere. We respectfully request that the syntax stay as-is because it does not seem to be a journal style guide requirement, but rather a personal preference. We are happy, however, to conform to the guidelines of the MDPI copyeditor.
- There are some typing mistakes as well, and authors are advised to carefully proofread the text. For example,
- all over the manuscript, the word “weightbearing” may be as “weight bearing”;
- in line number 42, “remain” as “remains”;
- in line number 61, “Mean time” as “Meantime”;
- in line number 234, “organism specific” as “organism-specific”;
- in line number 235, “broad spectrum” as “broad-spectrum”.
The typos not mentioned here are also to be checked and corrected properly.
Response: We must respectfully disagree about the recommendation at line 61, as “mean time” and “meantime” have distinctly different meanings; we are referring to an average as opposed to something that occurs simultaneously with something else. We acknowledge your recommendations at lines 42, 234, and 235 and have updated these accordingly in the manuscript. Lastly, we are familiar with the one-word phrase “weightbearing” in the literature (a PubMed search reveals 12,470 results) but agree to conform with this request and have updated it to two words throughout the manuscript.
- Check the abbreviations throughout the manuscript and introduce the abbreviation when the full word appears the first time in the text and then use only the abbreviation (For example, fracture-related infections (FRI), BMP-2, etc.). And it should be in both abstract as well as in the remaining part of the manuscript. Make a word abbreviated in the article that is repeated at least three times in the text, not all words need to be abbreviated.
Response: We found an instance of “fracture-related infections” at line 230 that we updated to FRIs. Otherwise, all seems OK as-is. We purposefully introduced RIA at line 77 and left that intact as we felt that readers may recognize the abbreviation than the definition (such as AIDS or EKG for example) but we are happy to conform to the guidelines of the MDPI copyeditor if they wish to remove the parenthetical abbreviation RIA and leave only our expanded definition.
- The full form of the species should be given when the first time appears in both abstract and in the remaining part of he manuscript and it should be followed by only the first letter of the genus (e.g., Staphylococcus aureus when the first time appears and followed by S. aureus). And it should be for all other species used.
Response: Thank you, Staphylococcus aureus is spelled out at new first appearance inserted at line 93. It has been updated to S. aureus at lines 139, 232 and 233, as well as in its second appearance in the the Table 4 legend at line 245.
- The introduction part appears less informative about the orthopaedic infection, thus this section should be indicated as detailed to understand the manuscript in clear.
Response: Thank you for this comment, we have revised it with this new text: Orthopaedic infections complicated by large soft tissue defects, exposed hardware and lack of bone stability have traditionally been treated with external fixation since inserting internal fixation into an infected bone and soft tissue envelope has been contraindicated. Bone stability is essential for eradicating infection. The concept of the insertion of antibiotic coated intramedullary nails has revolutionized the care of these difficult cases since it can provide antibiotic delivery and bone stability and be inserted at the time of radical infection debridement.
- In table 2, the words used may start with an upper case letter (capital letter).
Response: Thank you for pointing out this error, it has been updated in the manuscript.
- The authors may improve the discussion of their work by focusing on the present findings and introducing other authors who also worked with the same or other studies with recent references shortly.
Response: This was added at lines 143-147: In their retrospective study, Makhdom and his colleagues concluded that ACCINs represent an effective treatment option for in septic complex lower extremity reconstruction and associated with a limb salvage rate of 89%. They also recommended that patient with knee fusion after failed TKA should be counseled for potential high complication rate [3].
- In conclusion, it is highly recommended to include weaknesses of the study and potential future research goals.
Response: We have a limitations paragraph from lines 182-188: This study has limitations, including its retrospective nature by a single surgeon. Additional limitations are the absence of a comparative group, the variety of indications, and the presence of segmental bone defects in a subgroup of the series. Also the variation of the age group (range, 13 to 83 years) may have influenced the outcome. However, the authors posit that the variety of indications and the inclusion of segmental bone defects aid in showcasing the effectiveness of ACCINs in the treatment of bone infection in complex cases.
We added this at lines 319-321: Future multicenter randomized studies comparing the outcome of ACCINs vs other modalities in the treatment of bone infection are encouraged.
Reviewer 3 Report
The article by Conway et al. describes a retrospective series of patients who were treated with antibiotic-laden custom intramedullary nails. This is an important piece as it is the first study to describe such a large cohort. 111 patients were treated with antibiotic cement-coated intramedullary nails. Of these a positive outcome was obtained in 97 patients. In 67 of these patients the positive outcome was obtained after only 1 procedure, 30 patients required at least one additional procedure due to infection, nonunion or the combination of both. The authors therefore conclude that antibiotic cement-coated intramedullary nails can aid in the treatment of patients with FRI.
All patients were treated by the same surgeon, thereby decreasing the amount of variability between cases. Nevertheless, I have some remarks that should be addressed prior to publication. First of all, the terminology should be consistent throughout the manuscript. It is unclear what the distinction is between ‘osteomyelitis’, ‘fracture-related infection’ and ‘infected nonunion’. Also, the MSIS criteria for periprosthetic joint infection were used to diagnose infection in all patients (including FRI), while the FRI consensus definition has recently been validated. Please provide an overview of which diagnostic criteria were present for each indication, as a significant number of patients in the study cohort (42.3%) presented with negative cultures. Below I have listed more detailed comments for each section.
Results:
Line 51: 97 patients achieved either healed and uninfected bone or stable arthrodesis. What was the outcome of the remaining 14 patients? This is not immediately clear from the results section. Please give a more detailed overview.
Line 57: 30 patients required additional procedures. Would it be possible to provide an overview of culture results and antibiotic susceptibility for these patients?
In the case of patients with persisting infection or infection recurrence, were the isolated pathogens resistant to the used local antibiotics (vancomycin/tobramycin)?
Four of the 30 patients required additional procedures due only to nonunion, was tissue sampling performed in these cases? Which criteria were used to exclude an infectious process?
Table 1, Complications: 6 patients had a ‘superficial’ infection, based on which criteria was this diagnosed? Please describe in more detail. Were deep tissue cultures taken during the surgical debridement? How long were these patients treated with antibiotics?
Table 2, Rod removal: 7 patients required amputation, please provide some more detail on why these patients had to undergo amputation. Was this due to persisting infection? If so, which pathogens?
Discussion:
Line 122: Please print ‘Staphylococcus aureus’ in italic.
Lines 167-168: ‘This study demonstrates the effectiveness of the ACCIN technique in the setting of infected nonunions and arthrodeses’. It is not entirely clear how this can be concluded from the study, because there is no discussion on follow up cultures and the definitions/criteria that were used for nonunion, FRI, etc. Please explain this in more detail.
Methods:
Line 182, Table 3: The authors used the Cierny-Mader classification to classify patients’ host status. This classification specifically refers to patients with periprosthetic joint infections. However, for other patients (e.g. with FRI), this classification may be more difficult to extrapolate because it includes a time component that is arbitrary as well as a local extremity status that specifically refers to prosthetic joints. Since only 39% of patients were diagnosed with a PJI, was this considered?
Line 190: The abbreviation ‘TKAS’ likely refers to total knee arthroplasty, but is not defined in the manuscript.
Line 191, Table 3: It is not clear how the distinction is made between infected nonunion and FRI. Does infected nonunion only refer to an infected osteotomy? Which definition was used for nonunion? Please specify in the Methods section which (diagnostic) criteria were used for nonunion and infection. Was the consensus definition for FRI used to diagnose patients with this condition or was the MSIS definition for periprosthetic joint infection applied to all patients? Please specify this in the Methods section.
Table 4, Culture results: please provide a more detailed table caption.
- Please add species or spp. to the genera that are mentioned (e.g. Pseudomonas spp.) or mention the exact bacterial species when it is only one (e.g. Pseudomonas aeruginosa).
- Please correct to ‘Corynebacterium’
- Were all pathogens susceptible to either vancomycin or tobramycin?
- Were cultures also taken during the subsequent removal of the nail? Would it be possible to provide an overview of those cultures? Was there a correlation between bacterial species and outcome (e.g. more additional procedures due to infection recurrence)?
- A large number of patients had negative cultures (42.3%). How would you explain this? Were some patients under antibiotics at the time of sampling and which sampling protocol (e.g. number of tissue specimens) was used in these patients? Please also provide some more details about sampling protocols in the methods section.
Author Response
Reviewer #3
The article by Conway et al. describes a retrospective series of patients who were treated with antibiotic-laden custom intramedullary nails. This is an important piece as it is the first study to describe such a large cohort. 111 patients were treated with antibiotic cement-coated intramedullary nails. Of these a positive outcome was obtained in 97 patients. In 67 of these patients the positive outcome was obtained after only 1 procedure, 30 patients required at least one additional procedure due to infection, nonunion or the combination of both. The authors therefore conclude that antibiotic cement-coated intramedullary nails can aid in the treatment of patients with FRI.
All patients were treated by the same surgeon, thereby decreasing the amount of variability between cases. Nevertheless, I have some remarks that should be addressed prior to publication. First of all, the terminology should be consistent throughout the manuscript. It is unclear what the distinction is between ‘osteomyelitis’, ‘fracture-related infection’ and ‘infected nonunion’. Also, the MSIS criteria for periprosthetic joint infection were used to diagnose infection in all patients (including FRI), while the FRI consensus definition has recently been validated. Please provide an overview of which diagnostic criteria were present for each indication, as a significant number of patients in the study cohort (42.3%) presented with negative cultures. Below I have listed more detailed comments for each section.
Response: Concerning distinction of FRI vs infected nonunion, we did not distinguish between these for this study. This study was partially conducted prior to the FRI Consensus Meeting occurring. The senior author classified acute fractures less than 6 months and FRIs as those presenting after 6 months as infected nonunions.
Our criteria is present at lines 252-258 with some newly added text to this section and an additional reference: Infection was defined using the criteria from the Musculoskeletal Infection Society (with the presence of a draining sinus, erythema or abscess clinically, radiographically with osteolysis on radiographs or bone changes on MRI consistent with infection, positive intraoperative biopsies and elevated serologic markers [C-reactive protein and erythrocyte sedimentation rate]) in case of PJI [23]. In case of FRI, we had followed the same recommendations that were published recently to diagnose FRI [24].
- Parvizi, J.; Tan, T.L.; Goswami, K.; Higuera, C.; Della Valle, C.; Chen, A.F.; Shohat, N. The 2018 definition of periprosthetic hip and knee infection: An evidence-based and validated criteria. J Arthroplasty. 2018, 33, 1309-1314.e2.
- Govaert, G.A.M.; Kuehl, R.; Atkins, B.L.; Trampuz, A.; Morgenstern, M.; Obremskey, W.T.; Verhofstad, M.H.J.; McNally, M.A.; Metsemakers, W.-J.; Fracture-Related Infection (FRI) Consensus Group. Diagnosing fracture-related infection: Current concepts and recommendations. J Orthop Trauma. 2020, 34, 8-17.
Results:
Line 51: 97 patients achieved either healed and uninfected bone or stable arthrodesis. What was the outcome of the remaining 14 patients? This is not immediately clear from the results section. Please give a more detailed overview.
Response: We added this sentence at lines 57-59: For the remaining 14 patients, seven ended with amputation and seven were still undergoing treatment: three for infected nonunion and four for nonunion only after infection resolution.
Line 57: 30 patients required additional procedures. Would it be possible to provide an overview of culture results and antibiotic susceptibility for these patients?
Response: We added this at lines 60-68: Thirty patients underwent additional procedures. Fourteen of them had the same culture results postoperatively, and the remaining 16 had different results postoperatively. Twenty patients had positive culture results and the remaining 10 had negative culture results. Regarding antibiotics susceptibility for the 20 patients with positive culture results, 18 patients were sensitive to several antibiotics including vancomycin or tobramycin. One patient (with multiple organisms) was only sensitive to meropenem and was resistant to both tobramycin and vancomycin. The patient received meropenem postoperative for six weeks. The final patient of this group of 20 did not have antibiotic susceptibility information in the record.
In the case of patients with persisting infection or infection recurrence, were the isolated pathogens resistant to the used local antibiotics (vancomycin/tobramycin)?
Response: We added a sentence and new reference at lines 276-277: The combination of local vancomycin and tobramycin is favored in the treatment of osteomyelitis [25].
- Pargas CD, Elhessy AH, Abouei M, Gesheff MG, Conway JD. Tobramycin Blood Levels after Local Antibiotic Treatment of Bone and Soft Tissue Infection. Antibiotics (Basel). 2022;11(3):336. Published 2022 Mar 4. doi:10.3390/antibiotics11030336
Four of the 30 patients required additional procedures due only to nonunion, was tissue sampling performed in these cases? Which criteria were used to exclude an infectious process?
Response: Tissue samples were collected at every surgery and numbered between 3-5. Criteria were used to exclude an infectious process were the same all the way through our study.
Starting at line 287 is this information: All patients were monitored for resolution of infection and bone healing. Serial C-reactive protein and erythrocyte sedimentation rate values were obtained weekly for 6 weeks, followed by testing every 2 weeks to ensure that lab values returned to normal and stayed normalized following completion of 6 weeks of antibiotic therapy. Clinical monitoring of infection was performed routinely at 2, 4, 8, and 12 weeks postoperatively.
Table 1, Complications: 6 patients had a ‘superficial’ infection, based on which criteria was this diagnosed? Please describe in more detail. Were deep tissue cultures taken during the surgical debridement? How long were these patients treated with antibiotics?
Response: Superficial infection happened in the first 2 weeks after surgery. All patients were already on postoperative antibiotic treatment for a total of 6 weeks. This was added at lines 104-106.
Starting at line 273 is this text: Postoperatively, 6 weeks of organism-specific antibiotics were given. In culture-negative infections, broad-spectrum antibiotics were used. When possible in the culture-negative infections, historical cultures were used to guide antibiotic therapy. Antibiotic management was directed by the infectious disease consultant. Added at line 104: Superficial infections were handled with local wound care and without a return to the operation room.
Table 2, Rod removal: 7 patients required amputation, please provide some more detail on why these patients had to undergo amputation. Was this due to persisting infection? If so, which pathogens?
Response: We added this text at line 92: Four amputation patients had negative cultures and three were positive for Staphylococcus aureus. Five amputations were performed due to persistent infection and two were due to nonunion.
Discussion:
Line 122: Please print ‘Staphylococcus aureus’ in italic.
Response: Thank you for pointing out this oversight, it has been updated in the manuscript.
Lines 167-168: ‘This study demonstrates the effectiveness of the ACCIN technique in the setting of infected nonunions and arthrodeses’. It is not entirely clear how this can be concluded from the study, because there is no discussion on follow up cultures and the definitions/criteria that were used for nonunion, FRI, etc. Please explain this in more detail.
Response: At a mean of 29.2 months, follow-up 87.4% of patients were clinically, radiographically, and seriologically free of infection, demonstrating the effectiveness of the technique for indications of infected arthrodeses and FRIs. This is stated at lines 51-53, albeit with different wording, so we have now adapted it. Thank you.
Methods:
Line 182, Table 3: The authors used the Cierny-Mader classification to classify patients’ host status. This classification specifically refers to patients with periprosthetic joint infections. However, for other patients (e.g. with FRI), this classification may be more difficult to extrapolate because it includes a time component that is arbitrary as well as a local extremity status that specifically refers to prosthetic joints. Since only 39% of patients were diagnosed with a PJI, was this considered?
Response: We agree with this point, we used only the part of the CM classification for the Host type, which was used previously in several studies to describe the Host type based on several factors. We are planning to use other Host classifications in our future studies, like Charlson Comorbidity Index [Conway J, Mansour J, Kotze K, Specht S, Shabtai L. Antibiotic cement-coated rods: an effective treatment for infected long bones and prosthetic joint nonunions. Bone Joint J. 2014;96-B(10):1349-1354].
Line 190: The abbreviation ‘TKAS’ likely refers to total knee arthroplasty, but is not defined in the manuscript.
Response: Thank you for catching this, we have updated it to TKAs with a lower-case s.
Line 191, Table 3: It is not clear how the distinction is made between infected nonunion and FRI. Does infected nonunion only refer to an infected osteotomy? Which definition was used for nonunion? Please specify in the Methods section which (diagnostic) criteria were used for nonunion and infection. Was the consensus definition for FRI used to diagnose patients with this condition or was the MSIS definition for periprosthetic joint infection applied to all patients? Please specify this in the Methods section.
Response: We have added this text at lines 247-252: Although FRI is a challenging musculoskeletal complication in trauma surgery, it long lacked a clear definition. In their efforts to provide a clear definition for FRI, Metsemakers et al suggested that although acute vs chronic infections mandate different treatment strategies, this should not impact how clinicians define FRI [8]. In our series, the senior author classified FRIs as acute fractures <6 months and infected nonunions as those presenting >6 months.
Table 4, Culture results: please provide a more detailed table caption.
Response: Reviewer #1 suggested updating the caption to “Microbiological Cultures” and we have updated the manuscript accordingly.
- Please add species or spp. to the genera that are mentioned (e.g. Pseudomonas spp.) or mention the exact bacterial species when it is only one (e.g. Pseudomonas aeruginosa).
Response: We have also updated the names of the microbials.
- Please correct to ‘Corynebacterium’
Response: Thank you for pointing out this error, it has been updated in the manuscript.
- Were all pathogens susceptible to either vancomycin or tobramycin?
Response: We responded earlier to the previous reviewer with the same query: We added a sentence and new reference at lines 276-277: The combination of local vancomycin and tobramycin is favored in the treatment of osteomyelitis [25].
- Pargas CD, Elhessy AH, Abouei M, Gesheff MG, Conway JD. Tobramycin Blood Levels after Local Antibiotic Treatment of Bone and Soft Tissue Infection. Antibiotics (Basel). 2022;11(3):336. Published 2022 Mar 4. doi:10.3390/antibiotics11030336
And this text is at line 273: Postoperatively, 6 weeks of organism-specific antibiotics were given. In culture-negative infections, broad-spectrum antibiotics were used. When possible in the culture-negative infections, historical cultures were used to guide antibiotic therapy. Antibiotic management was directed by the infectious disease consultant.
- Were cultures also taken during the subsequent removal of the nail? Would it be possible to provide an overview of those cultures? Was there a correlation between bacterial species and outcome (e.g. more additional procedures due to infection recurrence)?
Response: Not recorded
- A large number of patients had negative cultures (42.3%). How would you explain this? Were some patients under antibiotics at the time of sampling and which sampling protocol (e.g. number of tissue specimens) was used in these patients? Please also provide some more details about sampling protocols in the methods section.
Response: Added at lines 233-240: Cultures were obtained for the four patients with nonunion only; their results were negative. The same criteria were followed for these 4 to confirm there was no infection and the revision was only for nonunion. Negative culture results can be associated with osteomyelitis, thus diagnosis involves clinical, laboratory, and radiological investigations. Some studies have suggested that the rate of negative cultures obtained during surgery in histologically proven osteo-myelitis exceeds 40% [21].
- Wu, J.S.; Gorbachova, T.; Morrison, W.B.; Haims, A.H. Imaging-guided bone biopsy for osteomyelitis: Are there factors associated with positive or negative cultures? AJR Am K Roentgenol. 2007, 188, 1529-1534.
Additionally added at line 240: Antibiotics were not routinely discontinued prior to surgery and could account for our culture rate. However, up to 30% negative culture rate has been reported in the literature [22].
- Tan, T.L.; Kheir, M.M; Shohat, N.; Tan, D.D.; Kheir, M.; Chen, C.; Parvizi, J. Culture-negative periprosthetic joint infection: An update on what to expect. JBJS Open Access. 2018, 3, e0060.
Round 2
Reviewer 3 Report
Dear authors,
Thank you for updating your manuscript.
I agree with the added extra sentence in the Methods section regarding the distinction of FRI vs. infected nonunion, as the pathogenesis of both entities is essentially the same: i.e., presence of bacteria surrounding the fracture. This is not influenced by the time component. Furthermore, the 6 months cut-off for non-union may be a bit arbitrary as for instance the FDA criteria for nonunion include the presence of a fracture for at least 9 months with at least 3 consecutive months without signs of healing. Please consider to add an extra sentence on this as well.
Author Response
Hello, we have modified the sentence to include your point. We had this sentence: In our series, the senior author classified FRIs as acute fractures <6 months and infected nonunions as those presenting >6 months. We have modified it to read: In our series, the senior author classified FRIs as acute fractures <6 months and infected nonunions as those presenting >6 months, whereas the US Food and Drug Administrations’s classification of a nonunion presenting ≥9 months without showing healing for 3 consecutive months.
We hope this resolves the issue you raised,
